# Parents’ Experience in an Italian NICU Implementing NIDCAP-Based Care: A Qualitative Study

**DOI:** 10.3390/children9121917

**Published:** 2022-12-07

**Authors:** Natascia Bertoncelli, Licia Lugli, Luca Bedetti, Laura Lucaccioni, Arianna Bianchini, Alessandra Boncompagni, Federica Cipolli, Anna Cinzia Cosimo, Giovanna Cuomo, Michela Di Giuseppe, Tamara Lelli, Veronica Muzzi, Martina Paglia, Lucia Pezzuti, Claudia Sabbioni, Francesca Salzone, Maria Cristina Sorgente, Fabrizio Ferrari, Alberto Berardi

**Affiliations:** 1Neonatal Intensive Care Unit, Department of Medical and Surgical Sciences of Mothers, Children and Adults, University Hospital of Modena, 41125 Modena, Italy; 2PhD Program in Clinical and Experimental Medicine, University of Modena and Reggio Emilia, 41125 Modena, Italy; 3Pediatric Unit, Department of Medical and Surgical Sciences of Mothers, Children and Adults, University of Modena and Reggio Emilia, 41125 Modena, Italy

**Keywords:** preterm infant, parents, NIDCAP, family-centered care, qualitative research

## Abstract

**Background**: The birth of a preterm infant and his/her immediate admittance to the Neonatal Intensive Care Unit (NICU) are sudden, unexpected, stressful and painful events for parents. In the last decade, in response to the increased awareness of the stressful experiences of parents, much attention has been paid to Family-Centered Care (FCC) and the implementation of the Newborn Individualized Developmental Care and Assessment Program (NIDCAP). According to the NIDCAP model, the infant–parents’ dyad is the core of the care provided by the NICU professionals to reduce the stress experienced by parents. So far, the literature does not show a clear correlation between parental experiences and the NICU practices according NIDCAP principles. **Aims**: To explore how parents of preterm infants experienced the NIDCAP-based care from admission to discharge, in particular, their relationships with NICU professionals and with other parents, and the organization of the couple’s daily activities during this process. **Design**: Qualitative exploratory study. **Methods**: Twelve parents of preterm infants born between January 2018 and December 2020 at the NICU of Modena, with a gestational age at birth of less than 30 weeks and/or a birth weight of less than 1250 g, were recruited. Three couples had twins, and the total number of infants was 15. All infants were followed for up to 24 months post-term age (PTA) for neurological outcomes. Each couple was given a semi-structured online interview about their experience during their infant’s hospitalization in the NICU up to discharge. The interview was developed around three time points: birth, hospitalization and discharge. The data analysis was conducted according to the template analysis method. **Results**: The admission to the NICU was unexpected and extraordinary, and its impact was contained by the skilled staff who were capable of welcoming the parents and making them feel they were involved and active collaborators in the care of their infant. The emotional experience was compared to being in a blender; they were overwhelmed by changing emotions, ranging from terrible fear to extreme joy. The couple’s activities of daily life were reorganized after the infant’s birth and admission to the NICU. Fathers felt unbalanced and alone in taking care of their partners and their children. **Conclusions**: This is the first study in Italy to explore parental experience in an NICU implementing NIDCAP-based care. The NIDCAP approach in the NICU of Modena helps parents to be involved early, to develop parental skills, and to be prepared for the transition home; and it also facilitates and enhances the relationship between parents and NICU staff.

## 1. Introduction

The birth of a preterm baby is always an unexpected and traumatic event for parents who are generally not well prepared for the extreme stress and wide range of emotions arising from admission to the Neonatal Intensive Care Unit (NICU). The main sources of stress for parents of a preterm infant are the fear for their baby’s health, changes in their parental roles, the baby’s behavior and appearance, and the highly technological environment of the NICU [1,2]. Obeidat et al. (2009), through a systematic review of qualitative studies, described the emotional distress of parents expressed through a sense of loss of control, conflicting feelings of hope and despair, and, among mothers, feelings of guilt for not being able to carry the pregnancy to term [3]. The need for information and reassurance about the baby’s clinical condition, and the desire to be close to their baby and to be able to participate in his or her caregiving process are the most important needs experienced by the parents during their baby’s stay in the NICU [4,5,6]. Words such as “up and down” and “roller coaster” are often used by parents to describe their experience in the NICU [7]. In response to the increased awareness of parents’ stressful experiences, many NICUs around the world, and also in Italy, have integrated the Family-Centered Care (FCC) principles into their care routines. FCC is a care approach based on mutual collaboration between staff and parents, respect, the dignity of individuals, participation, and sharing. The experiences of parents in the NICU with FCC-based care are variable: they range from a sense of guilt and loneliness to a sense of gratitude for a new life despite the prematurity of their infant [7,8,9]. FCC is an overall worldwide approach supported and disseminated by international and national guidelines that each NICU multidisciplinary team adopts [10]. The Newborn Individualized Developmental Care and Assessment Program (NIDCAP) approach is based on the concept of newborn competences observed by naturalistic observations of the infant behavior before, during and after care procedures. It focuses on infant’s behavioral cues to enhance his/her strengths and reduce his/her vulnerabilities [11], and parents are the most important nurturer for the infant. The NIDCAP method is meant for any NICU professionals, it requires a 2/3 years of training to obtain the certificate of NIDCAP professional. NIDCAP is the key tool of the FCC because it facilitates the shift from traditional task-oriented care to care based on relationships. The NIDCAP method helps NICU professionals to recognize the developmental needs of preterm infants, to support parents in their relationship with their infant, and in the acquisition and improvement of their parental competences [12].

Since its existence, the NICU of Modena has based its care on the NIDCAP method and model to meet the unique neurodevelopmental needs of infants and parents from admission to discharge. As far as we are aware, no qualitative studies on parents’ experience in an Italian NICU with a NIDCAP-based care model exists. With this in mind, we set out to explore parents’ experience and perceptions of the NIDCAP-based care in the NICU of Modena. 

## 2. Materials and Methods

### 2.1. Aim

The aim of the study was to explore how parents of preterm infants perceived the NIDCAP-based care from their infant’s admission to his/her discharge—particularly, the relationships with NICU professionals and with other parents, and the organization of the couple’s daily activities during this process.

### 2.2. Design

This qualitative exploratory study was conducted at the NICU of the hospital Azienda Ospedaliera Universitaria of Modena, Italy. Since 2002, the NICU started to adopt the NIDCAP-based model for the care of the newborn infants and their parents with the certification of the first NIDCAP professional in the unit. Since then, 12 more professionals were certified by this method, and in 2013 Modena NICU became a NIDCAP training center. The unit offers 24 h access, 7 days a week, to parents.

### 2.3. Participants

Participants were recruited among parents whose infants were born between January 2018 and December 2020 with a gestational age at birth of less than 30 weeks and/or a birth weight of less than 1250 g, and hospitalized at the Modena NICU. Every year, about 20 extremely-low-birth-weight infants (ELBW) are admitted to the NICU of Modena. Inclusion criteria for recruitment were absence of brain lesions from the infant’s point of view, and Italian mother tongue couples from the parents’ point of view. Twenty mothers were called on the phone by the first author (N.B.) and were asked to participate in the study together with their husbands; eight declined due to personal reasons. Twelve parent couples (n = 12 mothers, n = 12 fathers) were included and gave their informed written consent. Three couples had twins, and the total number of infants was 15. Median gestational age at birth was 29 (+1 standard deviation), and median birth weight was 973 (+71 standard deviation). The duration of hospitalization ranged from 43 to 115 days. As for clinical complications during NICU stay, the 15 infants were all similar. 

Information about mothers and fathers who participated in the study are not provided for privacy reasons.

### 2.4. Data Collection

The method used for data collection was a semi-structured interview with both mother and father. Parents were interviewed between 18 and 24 months after discharge of their baby. The interview guide was initially constructed on the back of the two decades of experience and expertise of one of the researchers (N.B.) at the Modena NICU on the theme of parental representation in the NIDCAP care process. The interview was developed around three time points considered relevant and adequate for eliciting answers from parents: admission, hospitalization, and discharge Three main topics were derived from these three events: the parent–child relationship and their early involvement in the care of their infant during hospitalization in the NICU, the parent–NICU-staff relationship, and the relationship and organization of couple’s daily activities. The interview started with a brief warm-up, during which the parents briefly described the life and health of their infant. After that, the topics were introduced by open questions (Table 1). One researcher (N.B.) conducted all the interviews from May to July 2021. All interviews were performed online in Italian, lasted from 30 to 60 min and were video recorded and transcribed. The first author (N.B.) was the gatekeeper, and her privileged position could have beneficially influenced the process of sensitively approaching and recruiting the parents for the interviews, resulting in rich and detailed narratives.

### 2.5. Data Analysis

The data analysis was conducted according to the template analysis [13], which is a form of thematic analysis with a flexible approach as to the style and the format of the template produced. It is based more extensively on the development of themes where the richest data are found, according to the research question. Central to the technique is the development of a coding template based on a set of data, which is subsequently applied to further data, revised, and refined [14]. The analysis process began in autumn 2021 and was performed by the first author (N.B.). The template was built starting from the interview guide, and the three relevant time points were considered. The first analysis of parents’ narratives was carried out by identifying meaning units in the text, i.e., sentences, and paragraphs with properties and meanings useful for their interpretation [13,15]. These meaning units were subsequently labelled with codes, and 15 themes were identified. These themes were then further clustered into 8 sub-themes which made it possible to interpret and describe the experience of the parents of a preterm baby from birth to discharge in the NICU. Codes were removed if they occurred only once or were irrelevant to the aim of the study. 

The eight sub-themes were interpreted and discussed in three domains according to an emotional and an organizational point of view:-Relationship with the NICU;-Emotional experience related to the NIDCAP-based care;-Organization of daily activities of parents/family.

All the interviews were translated into English by the first author (N.B.). Then, they were back-translated into Italian and checked for consistency.

### 2.6. Ethics

The Ethics Committee of Area Vasta Nord Emilia Romagna approved the information provided, recruitment methods, and type of consent. The parents participated voluntarily, and they received verbal and written information about the aim of the study. They were also informed that they could withdraw their participation at any time. 

## 3. Results

Eight sub-themes were grouped, and they were described and interpreted in three domains: relationship with the NICU, emotional experience related to the NIDCAP-based care, and organization of daily activities of the parents/family (Table 2). 

### 3.1. Relationship with the NICU

#### 3.1.1. Communication of Preterm Birth 

All parents felt preterm birth was an unexpected and totally sudden event which abruptly disrupted ordinary life; they had no knowledge of the specifics of preterm birth, and they would have needed information about the risks of a preterm birth. Regardless of how early parents knew about the preterm birth of their baby, they had little awareness of what was going to happen. All the mothers of preterm infants also described preterm delivery as an event that starts a journey down an unknown and unpredictable path.

*“I remember that morning. I was writing a message to my boss saying I’m going to the Emergency. ‘I don’t know if I can get back in the afternoon; at most see you tomorrow.’ And instead, I never came back.”* (Mother 12).

#### 3.1.2. Relationship with the NICU Staff

All parents reported that NICU staff had an empathic and humane approach both in the first days after birth and during hospitalization. The NIDCAP-based care made parents feel welcomed, they developed a sense of belonging, they established a sort of complicity with some professionals, and the NICU became a second family for them which was difficult to leave at the time of discharge. Parents also emphasized the centrality of the care in the infant–parents dyad in their experience in the NICU, and in terms of parents’ access and engagement.

*“I really like that in the NICU when you enter you are not* [a surname]; *you are not* [surname]; *you are not* [surname], *but there is* [name of infant] [...] *The care is centered on the baby; there are no timetables* [...] *In the NICU you were the dad and mum of your infant, there were no different treatments of an economic nature; huge. The NICU is a world that works perfectly, I even arrived at 4 am and there were people welcoming me.”* (Father 1).

Parents reported having perceived a great organization and the feeling of being cared for by a team of professionals who work together for the well-being of their babies. Three parents reported a sense of gratitude for feeling supported by the NICU staff.

*“I felt gratitude for those people who helped me a lot, because it’s not easy; you also need that contact. The fact that as soon as a full-term baby is born, the nurse immediately helps show how to bath the baby, how to change the nappy, which I did not do* [...] *With [name of the baby] it was more complicated; not everyone had the patience to get there and teach you, but there were those who went beyond your expectations and I appreciated this a lot.”* (Mother 7).

However, some parents reported that sometimes they experienced fear and their nerves becoming fired up due to the technical and clinical terminology used by the NICU staff when informing them about their baby. 

*“She [nurse] pops up and says, ‘Well, at most we can discharge her home with oxygen,’ as if she said, ‘Let’s have a good coffee.”* (Father 2).

Five parents reported the willingness of doctors and nurses to give them information about their baby’s clinical condition and to repeat it several times to make sure that it is fully understood. They underlined that how the information is given to them, including emphasizing positive aspects or not, contributed to making them feel better or worse and more or less confident in dealing with everyday life in the NICU. 

*“We asked and always had an answer; a lot changed in that moment.”* (Father 4).

### 3.2. Emotional Experience Related to the NIDCAP-Based Care

#### 3.2.1. Reactions at Birth 

All parents reported that the speed of the preterm birth prevented the experience of even unpleasant sensations, and it resulted in such a high level of stress that memories were erased. One mother reported the following.

*“And lucky it was so unexpected because one cannot prepare for such a thing; it is not humanly possible to prepare for such a thing.”* (Mother 2).

A couple of parents referred that the preterm birth of their infant was a surreal experience; they did not feel a part of it, and they had overlapping memories. They felt as if they were observing it from the outside. Another couple reported that the speed and criticality of the preterm birth completely shifted their attention to the newborn, who was often in mortal danger, and the parents’ experience became of secondary importance for the health professionals because the infant’s life was at stake.

*“It seems that we were experiencing something out of our own world. I don’t know—as if we were out of this thing* [...] *That is, something that we could not even decide or be involved in because everything happened so quickly that, in an automatic way, which in any case for the health professional is a bit automatic; caesarean delivery, preterm babies—it is known that they are admitted to the NICU and lots of things happen; however, in my opinion, no one focuses on what the couple is really experiencing.”* (Mother 4).

All mothers experienced a sense of emptiness because they suddenly felt the absence of their infant’s movements in the womb; they also reported the desire to see their infant, and the fear that seeing him/her would be different from what they expected. Although more and more efforts are made to make the NICU environment suitable and familiar for parents, the imagined infant is very different from the real infant in the NICU. The emotional impact on first contact with the infant is strong, and for all parents, very shocking.

*“... And all of a sudden you find yourself without your babies in your womb and you don’t even have the possibility to see them, to touch them.* [The father] *was taking photos, he was going upstairs, he was telling me about* [...] *but you don’t have them with you anymore, you don’t see them, and so, yes, that was it* [...]. *When you realize that they would have been born soon, somehow you try to rationalize it. When you don’t have them, you cannot touch them.”* (Mother 11).

#### 3.2.2. Reactions during Hospitalization

All mothers said that giving birth to a preterm infant was a very challenging and extremely stressful experience, involving considerable emotional commitment. All parents said that they had to perform an act of trust and leave their infant’s new life in the hands of experts and technicians who would take care of them. 

*“And you have to do an act of trust, because you can’t rely on the decision of anyone; you really have to do an act of trust and it’s very difficult* [...] *When I told you about a daily stand-by routine it was because at a certain point you have to trust the people you are dealing with.”* (Mother 2).

All parents reported that they tried to recognize their infant’s behavior. They felt that interpreting their infant’s cues was a privileged way to get in touch with him/her, know each other, and enter into a relationship.

*“*[...] *Every day as soon as I got there, I felt the need to give him my hand and felt that he held me, that is, I felt that he needed this support. Even if he was small, I think he felt that contact* [...]. *And afterwards he let my finger go and after a while I was aware that he was looking for that contact again; he tried to move his hand and he was looking for my finger. According to me, he gave the first signs of reaction, that he wanted to react to that situation. I say things that may not have anything to do with it, but they were my sensations.”* (Father 8).

All parents experienced a whirlwind of extreme emotions—great emotional instability and uncertainty—and that they felt overwhelmed. Two parents described their daily routine in the NICU like being in a blender and on a swing. 


*“’When you get inside the NICU you’re like inside a blender.’ It’s so true. On the second day, when I arrived, we were welcomed by a doctor and a nurse, and they asked us, ‘What do you know about the NICU?’*


*And we answered, ‘Nothing.’ And then they start to explain; they tell you there are days… Here it’s like a swing—days that are going very well and other days that are not; this is a metaphor that has never been more adequate.”* (Mother 8).

Two parents reported that they had to adapt to a “live in the moment” situation and to a daily stand-by routine. In the stand-by condition, the word “stable” became recurrent when they were given information by the NICU staff, and they accepted the sparing and cautious way of being informed daily, at least in the first few weeks of life.

*“Be happy when you hear ‘stable.’ However, it is something you learn over time. You don’t get it right away, in fact.”* (Mother 2).

All parents experienced and shared the fear of “breaking” and hurting their infant even with light contact of their hands or even with a glance. They felt that their infant was extremely fragile and vulnerable, that he/she could be broken if not handled with the necessary care and attention. One couple was amazed at how easily the NICU staff moved and changed the position of their infant despite the so many wires and sensors.

*“I remember one thing that impressed me right away—that is, to see the ease with which they* [NICU staff] *maneuvered them* [the infants], *moved them, when they lifted them up to change the bed, the nest. I said, ‘but oh my God, how do they do it!’ It seemed to me that I would break her at any time with all those wires. I remember that at the beginning they made me very afraid, and I said, ‘but how can I take her in my hands with all those wires? What if I displace something, if I hurt her?’”* (Mother 10).

The first skin-to-skin contact (SSC) with the mother and/or the father provides and produces a safe and developmentally expected environment that fosters the attachment that was abruptly interrupted with preterm birth. The first SSC is usually experienced by mothers who are welcomed and invited by NICU staff to hold their infants. The first SSC contact was always a pleasant memory for all mothers because they felt close and in tune with their infant and had the perception that he/she enjoyed it.

*“I remember fondly the first skin-to-skin. It was beautiful. I even had the shape of his tiny ear on my chest. And he was even intubated when I picked him up; he had the tube* [...] *All the alarms sounded, and I was turning around. And when the nurse pulled him off my chest, he was always standing there with his hands on me; he wanted to be there.”* (Mother 7).

All parents felt the need to be accompanied and involved step by step in the care of their infant from the very first days because it helped to acquire parenting skills. One mother perceived an apparently normal procedure like nappy changing as an almost heroic achievement that is worth sharing and reporting.

*“I was the first to be involved, because I was hospitalized, and I was constantly there. I remember those nappies, they were super small... I thought, ‘Oh god, now?’ Instead, I changed my mind and I started... The first two* [nappies], *oh god let me see; ‘look at me,’ but then I didn’t think any more, even if... I waited for the nurse to ask me, ‘Do you want to change her?’ Ready! I’ll take care of it; yes, yes, yes, even with all those wires, one hand on one side, and the other hand on the other side.”* (Mother 9).

Despite the early involvement of parents in daily care of their infant, at the beginning, five parents felt like spectators, waiting to be involved. They also reported the feeling of being suspended. They did not feel they were the “owners” of their family because at night they went home alone without their infant, and they were waiting for the next day to be able to take care of him/her again.

*“But as long as they’re in the NICU and someone else is taking care of them, you always run the risk of feeling like a spectator, as if sometimes your babies are not really yours; so, the chance to do skin-to-skin contact makes you feel, ‘Okay she’s my baby; she’s mine. I have her for a while.’ I cannot keep her because anyway when you go home you leave them there; they stay there and that always makes you feel a little bit in between, not having them with you.”* (Mother 11).

Mothers felt guilty for not being able to carry the pregnancy to term and for not having given birth to the infant they imagined. The sense of guiltiness and not feeling like real mothers appeared and grew during the years after birth, to the point that even after several years, mothers continued to live with this feeling without having really overcome it. One mother reported the following.

*“*[...] *And I collapsed... It is not possible; it’s all my fault; the sense of guilt for the preterm birth of your baby cannot be overcome. I had a hard time picking him up; his father took him.”* (Mother 7).

#### 3.2.3. Experience with Other Parents in the NICU 

Having empathy for other parents in similar situations was a challenging emotional experience for parents. There were situations in which they found it difficult to be happy about their infant’s improvement because they were aware of the worries and pains of other families. On the positive side, parents found of some reassurance to compare themselves to parents whose infants were further along. Sooner or later their infant would succeed in reaching that step forward.

*“In my opinion, one of the most difficult things in the NICU is that you share your experience with other mothers who are living the same experience but in a completely different way because the clinical conditions of their infants are totally different. Then, you share a bit of the happiness and the sadness of other parents because you feel it, even if it’s not your daughter who has that problem or makes that progress, and you have to force yourself not to see other parents’ emotions or at least I forced myself not to.”* (Mother 2).

#### 3.2.4. Emotions at Discharge 

The discharge from the NICU was a very emotional moment for all the parents, who experienced strong and even conflicting emotions. The transition home was referred to as a kind of separation from the safe environment of the NICU to an unknown place. All parents felt lost, bewildered, and alone because they were no longer cared for by the skilled NICU staff, but at the same time, they were happy. At discharge, some parents performed a sort of cost–benefit assessment and underlined that the experience in the NICU could also have had useful and enriching aspects for them from a personal point of view. They reported that, despite the blender-like experience and the swinging of emotions, the NICU stay had some benefits because it helped them to downsize and simplify some aspects of their daily life and to face them from different perspectives.

*“When I left the NICU, I started to cry. I was happy but I was a bit lost, I was sorry to leave because I felt that I was leaving a nest; it was really a strange feeling because I was so happy to leave, and I said, ‘And now what should I do, and now how do we manage it by ourselves? And saying goodbye to everybody was very sad because it had become my routine; it had become a sort of second family for us; it was very strange to leave—beautiful but strange.”* (Mother 3).

### 3.3. Organization of Daily Activities of Parents/Family

#### 3.3.1. Father’s Role

All fathers felt shocked and not prepared to face the birth of their preterm infant, and coping strategies were suspended. At birth, their first concern was for their partner’s health and only after for their infant. Six fathers reported a feeling of loneliness, and of being alone in taking care both of their partners and their infants in an extremely delicate situation. They reacted to this by masking and/or filtering their feelings and tried to remain solid in communication with family and friends about what was happening.

*“In that moment, I found myself completely on my own having to inform her parents, and my mother. I had to manage the situation; I had to accept that I had become daddy of who knows what it was, because she was such a little being that I was even struggling to realize it.”* (Father 2).

All fathers reported the joy of becoming a father despite the concomitant worries. Eleven fathers were involved early in caring for their infant during hospitalization, and they reported positive emotions; five fathers experienced skin-to-skin contact and reported that they remembered those moments very well because they were wonderful.

*“It went really well. I always have in mind the three and a half hours of skin-to-skin that I did with my daughter. I managed to hold her; she never moved for three and a half hours; she was great. I remember it well because that was when I held her the longest. The others [SSC] were also beautiful; that one I remember especially because it was the longest, and I remember it well.”* (Father 11).

#### 3.3.2. Organization of the Daily Activities of Parents/Family during Hospitalization

Most of the parents reported to have reorganized their routine, both work and family, after the birth and admission of their baby to the NICU. All mothers interrupted their work to dedicate themselves entirely to their infants, while the fathers continued their work and therefore spent less time with the infants. All parents reported that everything changed after birth, because they became parents and their priorities necessarily shifted to their infants. 

*“Well... I tell you I could do it because he [the father] was there and I could do it; I had the need to stay in the NICU with my daughter, he had the need to continue to do the things he did before* [...] *Not all families can afford the fact that the mother stops working to stay in the NICU with her daughter; this is not taken for granted.”* (Mother 2).

Parents gradually adapted to the NICU environment as long as they knew its sounds, lights, and staff, and it became their routine. After discharge, this routine was interrupted, and some couples felt a sort of imbalance. Three parents reported that their relationship was challenged in the first few months after discharge; fights began, misunderstandings increased, and there was a risk of separation. However, having lived together through such a strong and dramatic experience in the NICU allowed them to overcome the crises and come out of it with more strengths. For three parents, having another child acted as a glue to maintain balance within the couple.

*“We started fighting so much as we never did before; however, we realized that I would make it. I physically gave birth to her, but psychologically it was a birth of both of us, a pregnancy carried on by both of us.* [...] *We fortified ourselves.”* (Mother 3).

The criteria of scientific validity were met in our research. Saturation was reached after 10 interviews and further data collection would not help to develop a deeper understanding of the study domains.

## 4. Discussion

This is the first study in Italy on the experience of parents of preterm infants in a NICU where the NIDCAP model of care is implemented. 

This study highlights how parents dealt with the emotions at birth, at NICU admission, during hospitalization, and at discharge. Three macro-themes were identified: relationship with the NICU, emotional experience and organization of the daily activities of the couple/family.

The relationship with the NICU began when parents were informed about the preterm birth of their infant. Regardless of when the parents received this information, preterm birth was a sudden and unexpected event. The need to be informed about the risks of preterm birth, any possible complications related to it and their infant’s path after birth emerged from the narratives. Loewenstein et al. (2021), in their integrative review of qualitative studies on parental experiences and perceptions in the NICU, described similar findings. Communication, and being informed about their infant’s health by the NICU staff, were imperative for the parents. In order to cope with concern for the prognosis, fears, and uncertainty related to decision making, parents tried to gather information about their infant’s health [16]. The NIDCAP model of care, implemented at the Modena NICU, allowed parents to perceive the approach of the NICU staff as empathic and human both in the first days after birth and during the hospitalization. Parents reported having felt welcomed, and that they developed a sense of belonging. Nelson et al. (2016) found similar results in their study exploring the unique meaning of the core elements of mothering a preterm infant receiving NIDCAP-care in a level III NICU. Their findings highlighted that mothers of preterm infants in the NICU setting praise NIDCAP-based care for the education and support it provides [17].

All the parents perceived the NICU staff as well organized and a strong team, and they had the feeling of being able to trust them. The present study differs from that carried out by Finlayson et al., which investigated mothers’ perceptions of very preterm (VP) infants with respect to FCC in the United Kingdom. From the interviews of mothers during their infants’ hospitalization, a global theme emerged, identified as “finding my place.” All mothers found it difficult to “find their place” within the technocratic and clinical environment of the NICU [18]. These results suggest that the application of FCC principles as the main model of care in NICU is not enough to help parents to feel welcomed and supported during their NICU staying. The NIDCAP-based care at the Modena NICU is the key model to promote and facilitate a common approach of the staff in welcoming, supporting, and involving the parents. The NIDCAP model of care enhances the relationship among the triad and with the staff. 

Some parents reported a sense of gratitude for feeling supported by the NICU staff. They emphasized that they learned how to downsize and simplify some aspects of everyday life and to approach them from different perspectives, and the importance of appreciating the little things. The results of this study confirm those of Janvier et al. [19]. In their survey on the feelings of parents who were health professionals, working in the NICU or not, they highlighted that the sense of gratitude was common among all parents: gratitude for the gift of life, for simple and important things, for families and friends, for skilled professionals, and for the kindness of strangers. Parents’ emotions were closely related to their infants’ clinical conditions, which often changed rapidly and many times. Some parents described their NICU experience as “being in a blender” and as “a swing” of emotions that quickly and abruptly changed and followed one another. All parents described that they had to adapt to a “live in the moment” situation, a here-and-now condition, and to a stand-by daily life where the oscillations between steps forward and backward in their infant’s condition were common. These results are in line with the findings of Watson [6]. From the parents’ narratives obtained through semi-structured interviews, and a focus group, he identified the following common themes: fear for the life of the infant; uncertainty due to the unpredictability of the infant’s conditions, and impotence, due to the attitudes of some nurses who made parents feel like guests in the NICU. These results are also in line with those of a systematic review that investigated the experiences, behavior, and perceptions of parents of infants admitted to the NICU. The analysis of the nine selected qualitative studies highlighted three themes: stress due to hospitalization, alterations in parenting role, and the impacts of infant hospitalization on the psychological and emotional well-being of the parents [7]. The NICU was initially experienced in an oppressive way because the parents did not know it, they had never visited it, and almost everyone saw their babies there for the first time. Some parents were nervous and fearful, and others anxious and excited thinking of the first encounter with their baby. Despite the conflicting emotions experienced by parents at first contact, being able to physically touch their infants and becoming increasingly skilled and experienced in caring for him/her were important in triggering and/or strengthening their bonding. These results confirm those of some studies on early parent–infant contact [20]. The principles of Kangaroo Care (KC) support the importance of touch and physical contact with the infant as a nurturing approach; skin-to-skin contact is one of the fundamental elements of KC. Although very preterm infants are very sensitive to contact and repeated stimuli, it is proven that the gentle and comforting touch of parents, regulated with behavioral signs of their infant, is always beneficial and has positive value for both the infant and the parents [21]. The role of the father was a recurring theme in the narratives. Preterm birth is a shocking and unexpected event for fathers; they were not prepared, and they were not ready to face it. Emotional reactions oscillated between an initial blackout in response to the unexpected preterm birth to the joy of having become parents. At the beginning, each father’s needs were mainly informative and focused on seeking reassurance on the health of his partner and infant. The main coping strategies that fathers adopted were attempts to control and avoid negative feelings and to engage in the activities of daily life; returning to work was a kind of relief because fathers focused on things other than the NICU stress, and at the same time they felt useful to the new family. Fathers also reported being afraid of hurting their infants by touching them, despite their great will and desire to have physical contact with them. The involvement of fathers in the care of their babies increased during hospitalization; fathers were supported and encouraged to actively take care of their infants through SSC and daily care practices, such as nappy changes, feeding, and bathing. Our results are in line with those of Provenzi et al. (2015), who conducted a systematic review of qualitative studies on the experiences of fathers of very preterm (VP) infants admitted to the NICU, in terms of emotions, feelings, fears, and needs [22]. The results of this systematic review highlighted that fathers of preterm infants have a multidimensional emotional, cognitive, and behavioral experience of preterm birth and hospitalization in the NICU. As discharge approached, fathers reported that their emotions, needs, coping strategies, and care actions were continually redefined. 

Mothers often described a sense of guilt for their body that failed, not having been able to reach the due time of delivery, and of being incapable of protecting their infants in such a delicate developmental step. These results confirm those reported in a systematic review of qualitative studies on the parental experience in NICU. This review listed the many factors that increased stress levels related to admission of preterm infants to the NICU, including parental shame, guilt and social stigma, changes in family dynamics, and altered parenting roles [7]. Parents also identified their experience in the NICU as very intimate, yet empathetic at the same time. Having empathy for other parents in similar situations was identified by some parents as meaningful, with double significance. On the one hand, it was comforting and reassuring to know that the progress of the other parents’ infants could be the progress of their own; on the other hand, it was a sort of emotional overload because witnessing the backward steps of other infants and parents weighed down an already very demanding and intimate experience. Bry et al. collected similar results. They reported that the emotional social support that parents perceived from each other within the NICU was precious—sometimes even indispensable—because they lived and shared a common experience [23].

In this study, pairs of parents were interviewed within a couple of years after their infant’s discharge. Some memories of the parental experience may not be so vivid in the memory. On the positive side, having taken the time to reflect on their experience in the NICU may have helped parents to resolve the most important aspects of it and articulate their needs. It should be noted that being interviewed after discharge may have made parents feel freer to deliberately tell of their experience with respect to the NICU. 

No data were available about the social, economic, and family structures of these twelve parents. This lack of information could have been a potential confounder which was not considered in this study.

## 5. Conclusions

This study contributes to the understanding of parents’ experiences with their preterm infants in a NICU with a NIDCAP-based model of care. The admission, the hospitalization and the discharge seem to be stressful and shocking for parents because they are not prepared for the whirlwind of emotions and events into which they are drawn from the first moments after birth of their preterm infant. On the positive side, parents also experienced a sense of gratitude, they felt welcomed and supported, and the NICU staff became a sort of second family thanks to their skills in the relationship-based care according to the NIDCAP model.

## Figures and Tables

**Table 1 children-09-01917-t001:** Interview guide.

**1. Admission** 1.1 When and how did you know that your infant would be born before the expected date? 1.2 When and how did you get to know your infant for the first time in the NICU?
**2. Hospitalization** 2.1 Do you remember how and who explained to you how to hold him/her (skin-to-skin contact) for the first time? 2.2 Do you remember how and who taught you to bottle feed and/or breastfeed him/her? 2.3 How did the activities of daily life change within your couple during your infant’s hospitalisation? 2.4 What was the relationship with the parents of the other hospitalized infants?
**3. Discharge** 3.1 When and how were you told you were going home? 3.2 What would you like to forget or remember about the experience with your infant in the NICU?

**Table 2 children-09-01917-t002:** Sub-themes exemplars.

Domains	Emergent Themes	Exemplar
Relationship with the NICU	Communication of preterm birth	(F2) *“So, at that moment I would have preferred that they had explained it to me. In the obstetric unit there was a very nice doctor who came to talk to me as if my baby had little hope of surviving; I did not know anything about preterm birth. And she said to me, ‘Well ... if it is born, ... if it is born…’’If it is born, tell me: what should I prepare for?’ ‘If it is born, there is a unit.’ Ah well then, if there is a unit that takes care of these things, I’d rather resign myself.”*
	Relationship with the NICU staff	(F3) *“I used to say it; what those guys* [NICU staff] *do is 10% support for babies, 90% support for parents because I realize that a new parent who is facing such an experience asks even the most stupid and silly questions, and they probably answer those questions 100 times a day and they have to do it with a smile, reassuring us because a question that seems stupid to them helps a parent to be calm and not worried; so I realize that their work is not limited to taking care of the babies; rather, taking care of the babies is a small part; there is a lot of psychology behind it.”*
Emotional experience related to the NIDCAP-based care	Reactions at birth	(M7) *“The doctor comes to me and tells me, ‘Are you aware that your infant may not make it?’ ‘How can he not make it?’ Then I remember his leg. I saw only that one, from my position I saw only that one leg moving and I said, ‘He is alive.”*
	Reactions during hospitalization	(M12) *“*[...] *It’s difficult for me to explain it... In the sense that the feeling of being a mother, of becoming a mother has matured, but maybe it happens also with healthy baby, it takes time to develop because, maybe, when I was at home, I felt more like an unexperienced nurse.”*
	Experience with other parents in the NICU	(M11) *“*[...] *They (other parents) were our reference. We looked at them and thought, ‘They are now in the next room; in a few days it will be our turn too. They (other parents) are now in low-intensive-care; in a few days it will be our turn too; so, they were sort of our reference.”*
	Emotions at discharge	(M4) *“Anyway, it’s an experience (staying in the NICU) that makes you grow too... so much... and it is an experience that makes you understand also the importance of small things.”*
Organization of daily activities of parents/family	Father’s Role	(F4) *“I was always there, and while she was in the operating room in the afternoon, I was in the next room and I couldn’t wait for someone to come out, and tell me how she was, and* [...] *waiting for someone to come and comfort me that everything went well was very long* […] *I waited outside the NICU for about 4 h.”*
	Organization of the daily activities of parents/family during hospitalization	(F7) *“The one who kept us together was her sister. We didn’t want to burden her too much; we wanted to give her a sort of normality; we had to maintain a certain restraint, certain behavior in order to give her normality, even if we were traveling on parallel tracks, on two different tracks.”*

## Data Availability

Data are available upon request to the corresponding authors.

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
