# Peer review of "Parents’ Experience in an Italian NICU Implementing NIDCAP-Based Care: A Qualitative Study"

_children, 2022, doi:10.3390/children9121917_

Round 1
Reviewer 1 Report
Dear authors,
the study I had the chance to review seems interesting important. Athough the approach nor the idea is not original it is important to have a deeper look into the feelings of parents of pre-term babies in NICUs in different environments.
However, I have several major and minor concerns that prevent me from recommend this study to be published in Children without providing further explanation and improvements.
1) The methods of recruiting parents into the study are not clear and adequately described. It is not possible to consider possible biases and confounders if there is no information about the backgrounds: e.g. how many parents were approached, what was the response rate or at least how many preterm babies are carried about in this specific NICU.
2) The number of parent dyads included in this study seems quite low (12 parents of 15 babies) for a two year period so this should be commented or further discussed including the possible biases.
3) There is not Conclusion section in the article. Therefore, the authors miss the opportunity of simplified analytical conclusions and further views to be included.
4) Although the themes are divided into several domaines, personally, I find the article little bit difficult to read as the feelings of parents and impressions of reasearchers sometimes overlap. In my opinion, it would help if the authors could include some sort of overview with thematic analysis within these domains with different examples as sometimes seen in qualitative research articles.
5) Might be caused by the translation from the native language and omission of some worlds but in some occassion it seems difficult to understand what is exactly meant by some cited parental phrases. (e.g. on lines 479-480 - what will she make it? or 468-471 - what is not taken for granted? what cannot other families afford? or 407-408 - have to force yourself not to see who? what is meant by it?)
Author Response
Dear Reviewer 1,
thank you for your comments that will improve our manuscript.
1) Method: we added data about the number of ELBW infants admitted at our NICUs every year. According to the number of ELBW cared for in our NICU every year, 12 parent couples were considered a methodological adequate number for our qualitative study. We better defined inclusion criteria from infants and parents' point of view. We described further the recruitment of parents' couple.
2) Number of parent dyads: every year about 20 ELBW are admitted to the NICU of Modena. We recruited infants with a birth weight below 1250 g with no brain lesions. This inclusion criteria accounts for the number (n=12 mothers; n=12 fathers) of the parent dyads.
3) Conclusions: we added the Conclusion to the paper.
4) Thank you for this comment. We agree that sometimes the feelings of the parents and the reflections of the authors overlap. We modified the results, and we added Table 2 with exemplars of sub-themes in each domain.
5) We changed the sentences according to the reviewer comments.
Reviewer 2 Report
Abstract:
Some parts of the article need to be revised. Some elements in the background lack precision - NIDCAP does not aim to reduce parental stress and the fact that there is no correlation between parental experience and NIDCAP practices does not justify a qualitative study on parents' experience with NIDCAP. In the methodology section, neurological monitoring of infants is not relevant (not related to the variables being measured). Instead, it would be better to describe in brief the NIDCAP clinical practices applied in the neonatal unit (to which the parents who participated in the study were exposed). The last sentence of the conclusion is not supported by the results obtained in the study. There is too much extrapolation in my opinion. Revise this sentence to highlight the impact of the results on clinical practice.
Introduction:
- The Familily-centered care approach and the NIDCAP program are defined in the introduction. However, it is not clear why the FCC approach is discussed when the article is about NIDCAP. These are two different approaches in my opinion. Please describe the relationship between the FCC approach and NIDCAP and the specifics of each approach (what unites and differentiates these two approaches). It is important to be clear about how FCC can complement NIDCAP.
- For this sentence, please specify the negative and positive elements: The experiences of parents in the NICU with FCC-based care are variable: both positive and negative7-9.
- In sentence 73-74 you mention that NIDCAP has been used since the beginning of the Moderna neonatal unit. In the next sentence, you suggest that NIDCAP is being implemented. Please clarify. Please also clarify how NIDCAP is applied. In addition, please specify if and how the FCC approach is applied.
Materials and Mathods:
- Again, in the paragraph on the study specifications, it says that the NIDCAP program has been in place for 20 years. Do you mean that it was implemented in 2002 and has been applied and updated since then? This sentence is confusing.
- In the paragraph on participants, the neurological follow-up of infants is mentioned. How does this relate to the purpose of the study and the concepts being studied? The neurological follow-up seems to be irrelevant to the article since the study focuses on the period when the child is hospitalized and this paragraph refers to the neurological follow-up after hospitalization. Also, why describe the children and not the parents who participated in the study? Exactly how many parents participated? This is not clear because it mentions 12 families and fathers and mothers, but it also mentions 12 parents. Please clarify.
- Regarding Table 1, why is it presented in the method section when it is about results? Why present the socio-demographic characteristics of the infants when it is the parents who are the population under study? In my opinion, it does not make sense not to present the characteristics of the parents with the characteristics of the infants.
- Table 2 is difficult to understand. Please include more explanation in the text and modify the table to make it easier for the reader to understand.
- Please specify how the criteria for scientific validity were met during the study.
Results:
- It would be nice to reorganize the results to avoid repetition of certain elements such as the unexpectedness of preterm birth which is discussed several times in the description of the dimensions.
- Line 183 refers to FCC according to the NIDCAP model. This is confusing because the two approaches ( FCC and NIDCAP) were defined separately in the introduction and no link was made between the two. In addition, the purpose of the study refers to the NIDCAP program, not the FCC approach. How does the FCC approach of the caregivers influence the parents' perception of NIDCAP? The results for each approach should be separated, since the allusion to FCC in the results presenting parents' experiences with NIDCAP is confusing to the reader.
Discussion:
- The results of the study are compared with the results of previous studies investigating the FCC approach. Since FCC and NIDCAP are two different elements, these results are not very comparable in my opinion.
- Please elaborate further on what the results of this study add to the scientific literature.
Author Response
Dear Reviewer 2,
thank you for your comments that will improve our manuscript.
Introduction: FCC is the individualized developmental and family-centred care approach to the care of preterm infants in the NICU; the NIDCAP method is based on the naturalistic observation of preterm infant behavior. This method led to adapt NICU care to infant's need in order to minimize pain and stress, and to involve parents in the care of their infant. NIDCAP is the key tool for the FCC and it requires a 2/3 years training of single professionals; FCC is an overall worldwide spread approach supported and disseminated by international and national guidelines that each NICU multidisciplinary team. We provided the required description of the relationship between FCC and NIDCAP.
Materials and method:
- Modena NICU started the NIDCAP-based care in 2002 with one NIDCAP certified professional (myself) in the method and up to now 12 Modena NICU professionals are certificated in the NIDCAP method, increasing and empowering the relationship-based care for infants and families in Modena NICU. We changed the paragraph to make it clearer for the readers.
- We deleted the paragraph on the follow up because we agree it is irrelevant to the aim of the study.
- We mentioned the number of mothers and fathers that participated to the study.
- We stated in the paper that we didn't provide information about the parents for privacy reasons. Some parents didn't give their consent to have their personal information described in the paper.
- We removed Table 1 (infants' clinical characteristics) from the paper. We introduced a new Table 1 that now contains the guide of the interview that we think it's more relevant and could help the readers.
- Scientific validity: Validity in qualitative research means “appropriateness” of the tools, processes, and data. In a qualitative study, sufficient sample is the guarantee of research validity, and saturation is an indicator used to assess the adequacy of research data. Saturation means that on the basis of the currently collected and analyzed data, further data collection will not help researchers develop a deeper understanding of the theory, so there is no need to continue to collect data. In our study, we reached data saturation after 10 interviews. The last 2 interviews confirmed the emergent sub-themes. We added scientific validity to our manuscript.
Results
- Thank your for this comment. We rewrote the results in order to avoid repetitions of similar concepts, and we introduced Table 2 which contains examplars of the results for sub-themes in each domains.
- According to your previous comment, in the introduction we clarified the correlation between FCC and the NIDCAP-based care. We also changed the results accordingly.
Discussion: we further elaborated the discussion according to the updated literature on this topic.
We added the Conclusion.
Round 2
Reviewer 2 Report
The majority of the issues raised have been adequately addressed. However, in my opinion, there are still two points that need to be addressed:
These elements were not changed in the abstract, so the abstract is inconsistent with the changes made in the rest of the manuscript: NIDCAP does not aim to reduce parental stress and the fact that there is no correlation between parental experience and NIDCAP practices does not justify a qualitative study on parents' experience with NIDCAP. In the methodology section, neurological monitoring of infants is not relevant (not related to the variables being measured). Instead, it would be better to describe in brief the NIDCAP clinical practices applied in the neonatal unit (to which the parents who participated in the study were exposed). The last sentence of the conclusion is not supported by the results obtained in the study. There is too much extrapolation in my opinion. Revise this sentence to highlight the impact of the results on clinical practice.
Please add details about the other scientificity criteria. Achieving data saturation is only one element of the scientific validity criteria. This part of the manuscript is incomplete in my opinion. I recommend that you refer to a known author who has addressed these criteria.